# The Inverse Correlation of Isoflavone Dietary Intake and Headache in Peri- and Post-Menopausal Women

**DOI:** 10.3390/nu14061226

**Published:** 2022-03-14

**Authors:** Mayuko Kazama, Masakazu Terauchi, Tamami Odai, Kiyoko Kato, Naoyuki Miyasaka

**Affiliations:** 1Department of Obstetrics and Gynecology, Tokyo Medical and Dental University, Tokyo 113-8510, Japan; 180202ms@tmd.ac.jp (M.K.); n.miyasaka.gyne@tmd.ac.jp (N.M.); 2Department of Women’s Health, Tokyo Medical and Dental University, Tokyo 113-8510, Japan; odycrm@tmd.ac.jp (T.O.); kiyo.crm@tmd.ac.jp (K.K.)

**Keywords:** menopause, isoflavone, daidzein, genistein, headache

## Abstract

This study investigated the relationship between headache and dietary consumption of a variety of nutrients in middle-aged women. This cross-sectional analysis used first-visit records of 405 women aged 40–59 years. The frequency of headaches was assessed using the Menopausal Health-Related Quality of Life Questionnaire. Of the 43 major nutrient intakes surveyed using the brief-type self-administered diet history questionnaire, those that were not shared between women with and without frequent headaches were selected. Multiple logistic regression analysis was used to identify nutrients independently associated with frequent headaches. After adjusting for background factors related to frequent headache (vasomotor, insomnia, anxiety, and depression symptoms), the estimated dietary intake of isoflavones (daidzein + genistein) (mg/1000 kcal/day) was negatively associated with frequent headaches (adjusted odds, 0.974; 95% confidence interval, 0.950–0.999). Moreover, the estimated isoflavone intake was not significantly associated with headache frequency in the premenopausal group, whereas it significantly correlated with that in the peri- and post-menopausal groups. Headache in peri- and post-menopausal women was inversely correlated with the dietary intake of isoflavones. Diets rich in isoflavones may improve headaches in middle-aged women.

## 1. Introduction

Headache is quite common among women and is one of the most prevalent symptoms of menopause [1]. Most menopausal symptoms checklists include headache, and in our previous cross-sectional study, headaches were the 13th most common menopausal symptom of the 21 categories, with 52.8% of middle-aged women complaining of headache at least once a week [2]. The two most common subtypes of headache are tension headaches and migraines. The former is reported to be unchanged or worsened after menopause in more than two-thirds of female patients [3]. Contrarily, as the latter can be triggered by estrogen withdrawal, symptoms tend to improve after menopause [4] although it is still reported in 11–24% of post-menopausal women [4].

Studies have shown that specific diets, such as a low-calorie diet, carbohydrate-restricted diet, and weight-loss diets for obese patients, can improve headaches [5]. In a randomized controlled trial of 182 patients, high n-3 fatty acid diets ameliorated the frequency and severity of headache compared to the control diet [6].

The effect of diet on fluctuating estrogen-related headaches has also been studied. Namely, vitamin E intake has been reported to improve menstrual migraine in a double-blind controlled trial [1]. However, there have been few reports on the relationship between diet or nutrients and headaches especially in women transitioning through menopause. This study aimed to investigate the relationship between headache and dietary nutrient intake in middle-aged Japanese women.

## 2. Materials and Methods

### 2.1. Study Population

In the present cross-sectional study, we used the first-visit records of 2090 Japanese women enrolled in the Systematic Health and Nutrition Education Program (SHNEP) conducted at the Menopause Clinic of Tokyo Medical and Dental University, Tokyo, Japan, between March 1995 and January 2021. All participants in the program visited or were referred to our clinic for the treatment of menopausal symptoms. Of these, 494 responded to the Menopausal Health-Related Quality of Life Questionnaire (MHR-QOL) [7] and the brief-type self-administered dietary history questionnaire (BDHQ) [8,9]. The 46 patients treated with hormone replacement therapy, 35 patients younger than 40 years or older than 59 years, and 4 patients with unknown menopausal status were excluded, leaving 409 patient records included for analysis.

The research protocol was reviewed and approved by the Tokyo Medical and Dental University Review Board (number: 774, approval date: 23 March 2010), and written informed consent was obtained from all participants. This study was conducted following the Declaration of Helsinki.

### 2.2. Menopausal Status

We classified the women into three menopausal statuses: premenopausal, perimenopausal, and postmenopausal. The premenopausal group had regular menstrual periods over the past three months, the perimenopausal group had experienced menstruation in the past 12 months but not in the past three months or had irregular menstrual cycles, and the postmenopausal group had no menstruation in the past 12 months.

### 2.3. Physical Assessment

Body composition variables, such as body mass index, body fat percentage, fat mass, lean mass, muscle mass, and basal metabolism, were measured using a bioimpedance analyzer (MC190-EM; Tanita, Tokyo, Japan). Height, weight, and waist and hip circumference were measured to calculate body mass index and the waist–hip ratio. Body temperature was measured using a thermometer. Resting metabolic rate was estimated from the respiratory volume using an indirect calorimeter (Metavine-N VMB-005 N; Vine, Tokyo, Japan). Additionally, cardiovascular parameters, including blood pressure, heart rate, cardio-ankle vascular index, and ankle-brachial pressure index, were measured using a vascular screening system (VS-1000; Fukuda Denshi, Tokyo, Japan). We also conducted a physical fitness test to assess power, reaction time, and flexibility. Hand-grip strength was measured twice with a hand dynamometer (Yagami, Nagoya, Japan), and the mean value (kgf) was calculated using the larger value of the two measurements. The test for reaction time was repeated seven times with the ruler-drop test using a wooden ruler (Yagami, Nagoya, Japan) with a length of 60 cm and a weight of 110 g. The ruler-drop test was carried out by allowing a seated participant to affix their arm on a desk, extending their hand from the edge, while the examiner holds out the ruler above the participant’s thumb and index finger. The participant then attempts to catch the ruler as quickly as possible when it drops. The locations a participant grasped the ruler for each trial were evaluated, and the average reaction time (cm) was calculated from the remaining five values omitting the maximum and minimum values. Flexibility was measured by forward bend test (cm) using a reach box while sitting (Yagami, Nagoya, Japan).

### 2.4. Lifestyle Characteristics

We surveyed lifestyle factors, including smoking history (none, less than 20 cigarettes per day, more than 20 cigarettes per day), alcohol consumption (never, sometimes, daily), caffeine consumption (never, 1–2 times per day, 3 or more times per day), and regular exercise habits (yes, no).

### 2.5. Questionnaires

The MHR-QOL, developed and validated in our clinic [7], is a modification of the Women’s Health Questionnaire and other questionnaires [10,11,12]. Physical and psychological symptoms were scored on a 4-point Likert scale based on the frequency of symptoms. In this analysis, the scores increased as each symptom became more frequent (0, zero to once a month; 1, one-two times per week; 2, three-four times per week; 3, almost every day). In the present study, the participants that answered “3” under headache were defined as having frequent headaches, while all others were defined as without frequent headaches. The sum of the scores under hot flashes and night sweats was used as the vasomotor symptom (VMS) score (0–6). Likewise, the scores under difficulty in initiating sleep and non-restorative sleep were pooled to define the insomnia symptom score (0–6).

The Hospital Anxiety and Depression Scale (HADS) is a widely used and reliable screening test for mental health in patients with somatic symptoms [13]. The HADS consists of seven items for two subscales, depression and anxiety symptoms, and participants respond to each item on a 4-point Likert scale. Patients who rated 8–10 were classified as likely to have anxiety or depression, while those who rated 11–21 were classified as having anxiety or depression.

The BDHQ is a self-administered questionnaire that assesses the participant’s intake of 61 food items commonly consumed in Japan. The participants answered 77 questions regarding their consumption over the previous month. Based on their responses, the intakes of 96 nutrients were estimated using an ad hoc computer algorithm. A semi-weight method was used to adjust for total calorie intake [8,9]. In this study, we investigated the association between the frequency of headaches and the estimated intake of 43 major nutrients.

### 2.6. Statistical Analysis

Continuous variables are presented as mean ± standard deviation (SD). The required sample size, determined to be 333, was derived by multiplying 10 with the number of independent variables, then dividing the product by the prevalence of frequent headaches (estimated to be 5 and 0.15, respectively). Differences between the groups were compared using an unpaired *t*-test, Mann–Whitney test, chi-square test, and Fisher’s exact test. Multicollinearity between variables was determined using the cutoff points for Pearson or Spearman correlation coefficients of |R| > 0.9. The nutrients and socio-demographic factors that significantly differed between the group with and without frequent headaches were selected. After adjustments for the selected socio-demographic factors, the association between the frequency of headaches and the selected nutrients was analyzed using a multivariate logistic regression model. Statistical significance was set at *p* < 0.05. All Statistical analyses were performed with GraphPad Prism version 9.1.2 (GraphPad Software, San Diego, CA, USA) and JMP version 14.0.2 (SAS Institute Inc., Cary, NC, USA).

## 3. Results

The participants’ average age was 50.1 ± 3.8 years (mean ± SD). The number (percentage) of women who scored the frequency of their headaches as 0 was 176 (43.0%), while 122 (29.8%) answered 1, 51 (12.5%) answered 2, and 60 (14.7%) answered 3, indicating that more than half of the participants suffered from a headache at least once a week. The socio-demographic factors that were significantly different between the two groups, with and without frequent headaches, are presented in Table 1. Participants with frequent headaches had higher VMS, insomnia, anxiety, and depression scores than those without. We also assessed the dietary intake of 43 major nutrients, which differed significantly between the two groups (Table 2). The estimated intakes of vitamin K, daidzein, and genistein were lower in women with frequent headaches than in those without. A test of multicollinearity for these variables showed that the correlation coefficient between daidzein and genistein was 1.00. Therefore, the estimated isoflavone intake was defined as the sum of daidzein and genistein. The values of the other correlation coefficients were less than 0.7. 

Afterwards, multivariate logistic regression analysis was performed to reveal an independent association between the intake of selected nutrients (vitamin K and isoflavones) and the frequency of headache (Table 3). After adjustment for the selected nutrients (Model 1) and the socio-demographic factors significantly related to the frequency of headache (Model 2), the estimated intake of isoflavone remained significantly associated with frequent headache (Model 1: adjusted odds ratio (OR) per mg/1000 kcal/day = 0.976, 95% confidence interval (CI) = 0.952–1.000, *p* = 0.048; Model 2: OR = 0.974, CI = 0.950–0.999, *p* = 0.036), while the intake of vitamin K was not. In Model 2, VMS and insomnia scores had a significant relationship with the frequency of headache (VMS: OR = 1.160, CI = 1.012–1.329, *p* = 0.033; insomnia: OR = 1.334, CI = 1.152–1.546, *p* < 0.001).

Furthermore, in order to investigate the effect of estrogen fluctuation, the participants were divided into two groups: premenopausal (*N* = 134) and peri- and post-menopausal (*N* = 271), and the relationship between the frequency of headache and the dietary intake of isoflavone was evaluated in each group (Figure 1). The premenopausal group showed no significant difference in the isoflavone intake (mg/1000 kcal/day) between the women with frequent headaches and those without (21.6 (SD, 16.2) vs. 21.3 (SD, 11.2), *p* = 0.391, Mann–Whitney test). Contrary, in the peri- and post-menopausal group, the participants suffering from frequent headaches were found to consume significantly less isoflavone than those without (20.3 (SD, 15.8) vs. 26.6 (SD, 15.6), *p* = 0.011, Mann–Whitney test). 

## 4. Discussion

In this cross-sectional study, the estimated dietary intake of isoflavone was inversely associated with frequent headaches in middle-aged Japanese women independent of VMS and insomnia symptom scores. The socio-demographic factors were found to be associated with headaches. A significant difference in isoflavone intake between those with and without headaches was observed in the peri- and post-menopausal groups but not in the premenopausal group. 

Isoflavones, including daidzein and genistein, are flavonoids that are especially abundant in leguminous plants. Isoflavones are classified as phytoestrogens because they are structurally similar to 17β-estradiol and have estrogenic and anti-estrogenic effects as well as antioxidant effects [14,15]. Isoflavones have been shown to improve various menopausal symptoms, presumably through an estrogenic effect [16,17]. Isoflavones bind to both estrogen receptors (ER), ERα and ERβ, with a higher affinity for the ERβ receptor [14]. Although the binding activity of isoflavones to the ERs is much weaker than that of estradiol [18], the concentration of serum isoflavone in people consuming soy foods is several orders of magnitude higher than the physiological concentration of estrogen [19], suggesting that isoflavones have a certain physiological effect despite their relatively low ER affinity.

Few reports inferred that isoflavone alleviates premenstrual headache syndrome [20]. In a randomized controlled trial (RCT) of 23 women, consumption of isolated soy protein containing isoflavone (68 mg/day) significantly reduced premenstrual headache [21]. However, to the best of our knowledge, no study has investigated the association between isoflavone intake and headaches in middle-aged women despite headaches being prevalent in women transitioning through menopause (10–29%) [4]. The results of the present study suggest that dietary intake of isoflavone could improve the headache experienced by middle-aged women.

Several pathways could be involved in the association between isoflavone intake and improvement in headache (Figure 2). It has been established that soy isoflavone relieves VMS, especially hot flashes [17,22,23]. The link between VMS and headaches is not clear, but the relationship between VMS and insomnia symptoms is evident [24]. Insomnia symptoms, along with depressive symptoms, improved with a low dose of isoflavone (25 mg/day) in our RCT that enrolled middle-aged women [25]. It is well-known that headaches and insomnia have a bidirectional relationship; headaches can contribute to sleep disturbances, while sleep disturbances can be a predictor or trigger for headache attacks [26]. The other pathways involve anxiety and depression symptoms, known to be associated with headaches. Patients with generalized anxiety disorder (GAD) were at a higher risk for migraine (OR 3.86, 95% CI 2.48 to 6) than those without; accordingly, patients with migraines were at a higher risk for GAD (OR 3.13, 95% CI 1.56 6.3) [27]. Likewise, depression in migraineurs is 2.5 times more common than that in non-migraineurs and is often associated with anxiety symptoms [27]. Several reports, including our recent RCT [25], have shown that isoflavone intake alleviated depressive symptoms in peri- and post-menopausal women [28]. These findings suggest that the relationship between headaches and isoflavone intake could be influenced by the pathways involving VMS and insomnia, depression, and anxiety symptoms. However, in this study, we have shown that isoflavone intake is related to headache independent from these symptoms, which suggests that isoflavone might have a direct effect on headache.

Concerning the mechanism by which isoflavone directly alleviates headache, both its antioxidant and estrogenic effects could be candidates. Considering that the estimated amount of isoflavone intake was different between those with and without headache in the peri- and post-menopausal groups but not in the premenopausal group, the plausible mechanism by which isoflavone reduces headache would be through its estrogenic effect rather than its antioxidant effect. Transitioning through menopause, serum estradiol levels in women fluctuate widely and then decrease [29], which is a trigger of migraine in peri- and post-menopausal women [3]. The estrogenic effect of isoflavone may compensate for the fluctuations and decrease of estrogen [30,31]. Estrogen is known for its regulation of neuronal excitability, interaction with the vascular endothelium in the brain, and association with neurotransmitters, such as serotonin and norepinephrine. [32]. In the cardiovascular and central nervous system, ERβ is predominant, to which isoflavones have a higher affinity [33]. These findings suggest that isoflavone could improve headaches in middle-aged women through its estrogenic effect.

One of the major limitations of this study is the relatively narrow population of research participants attending menopause clinics in Japan. Since the intake of soy, a rich source of isoflavone, varies widely from country to country [34], it would be difficult to generalize this results to a broader population. Additionally, the questionnaire only obtained the frequency of headaches without distinction between different types of headaches, such as migraine and tension headache. Since the pathogeneses of these headaches are different, the relationship between isoflavone intake and each subtype needs to be further investigated although there is a challenge in diagnosing the exact type of headache. Another limitation was the cross-sectional design of our current study, which did not allow us to determine the causal relationship between the estimated intake of isoflavone and headache. 

Nevertheless, this study has several strengths. The extensive socio-demographic factors related to headache were simultaneously investigated, including age, menopausal status, body composition, cardiovascular parameters, basal metabolism, physical fitness, lifestyle factors, vasomotor symptoms, and psychological symptoms, in addition to the estimated dietary intake of 43 major nutrients evaluated. As far as we know, this is the first report to show an association between dietary isoflavone intake and menopausal headache symptoms.

## 5. Conclusions

Headache in peri- and post-menopausal women was inversely associated with dietary intake of isoflavone. Diets rich in isoflavones might improve headaches in middle-aged women.

## Figures and Tables

**Figure 1 nutrients-14-01226-f001:**
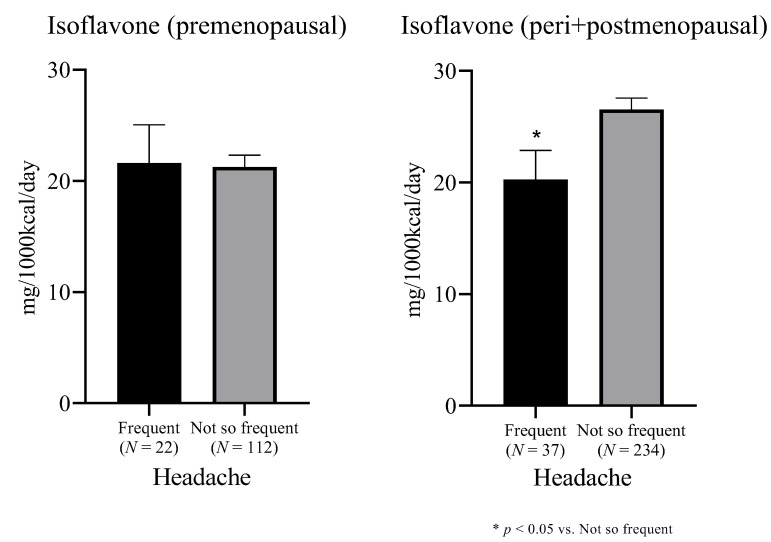
Comparison of daily isoflavone intake by menopausal status between women with frequent headaches and those without. * *p* < 0.05 vs. Not so frequent, Mann-Whitney test.

**Figure 2 nutrients-14-01226-f002:**
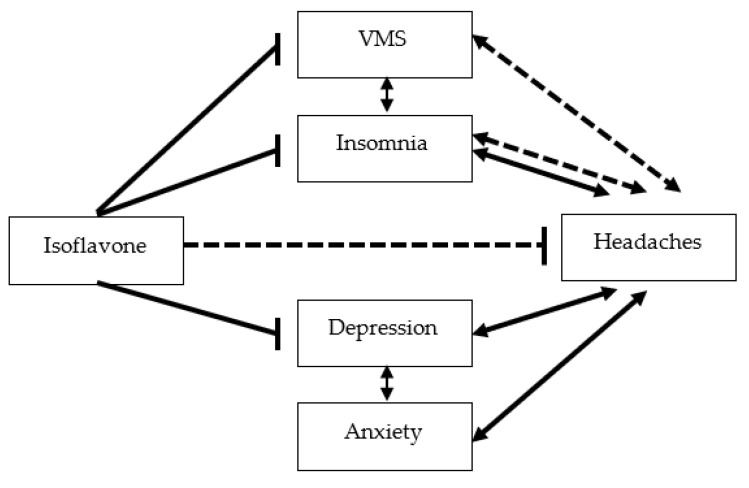
Possible pathways from isoflavone to headache. Arrows, bidirectional relationships; T-bars, inhibitory relationships; solid lines, relationships suggested from previous studies; dashed lines, relationships suggested in this study.

**Table 1 nutrients-14-01226-t001:** Comparison of socio-demographic factors and the frequency of headache.

Characteristic	Headache		*p*-Value
	Frequent (*N* = 60)	Not So Frequent (*N* = 349)	
Age and Menopausal Status			
Age (years), mean (SD)	49.4 (3.9)	50.2 (3.8)	0.128 ^a^
Menopausal status, premenopausal/peri-menopausal/postmenopausal (%)	37.3/27.1/35.6	32.4/23.1/44.5	0.441 ^b^
Body composition, mean (SD)			
Height (cm)	157.4 (5.0)	157.6 (5.4)	0.839 ^c^
Weight (kg)	54.0 (10.8)	54.4 (9.6)	0.691 ^a^
Body mass index (kg/m^2^)	21.8 (4.1)	21.9 (3.8)	0.616 ^a^
Waist–hip ratio	0.87 (0.07)	0.87 (0.06)	0.762 ^c^
Fat mass (kg)	15.9 (8.2)	15.8 (7.1)	0.899 ^a^
Lean mass (kg)	38.1 (3.3)	38.5 (3.5)	0.355 ^c^
Muscle mass (kg)	35.9 (3.1)	36.3 (3.2)	0.355 ^c^
Body fat percentage (%)	27.9 (8.8)	28.0 (7.7)	0.927 ^c^
Basal metabolism (MJ/day)	1107 (125)	1118 (122)	0.491 ^a^
Resting energy expenditure (MJ/day)	1664 (400)	1578 (428)	0.171 ^a^
Body temperature (℃)	36.2 (0.5)	36.3 (0.5)	0.765 ^a^
Physical fitness test, mean (SD)			
Hand-grip strength (kg)	25.2 (4.8)	25.3 (4.8)	0.834 ^c^
Ruler-drop test (cm)	22.9 (4.8)	22.8 (4.0)	0.895 ^a^
Sit-and-reach test (cm)	35.9 (10.0)	36.1 (9.7)	0.794 ^a^
Cardiovascular parameters, mean (SD)			
Systolic blood pressure (mmHg)	128 (16)	126.5 (17)	0.433 ^a^
Diastolic blood pressure (mmHg)	82.0 (12.5)	80.8 (12.2)	0.350 ^a^
Heart rate (per min)	65.0 (11.3)	63.7 (11.7)	0.441 ^a^
Cardio-ankle vascular index	7.47 (1.12)	7.42 (0.69)	0.148 ^a^
Ankle-brachial pressure index	1.11 (0.08)	1.10 (0.06)	0.077 ^c^
Lifestyle factors (%)			
Smoking (cigarettes per day)			
None/fewer than 20/20 or more	88.3/8.3/3.3	89.7/7.5/2.9	0.953 ^b^
Drinking			
Never/sometimes/daily	63.3/25.0/11.7	56.9/31.0/12.1	0.609 ^b^
Caffeine (per day)			
Never/1–2 times/3 or more times	13.3/36.7/50.0	6.9/34.5/58.6	0.179 ^b^
Exercise			
Moderate exercise Yes/no	51.7/48.3	43.0/57.0	0.260 ^d^
Regular exercise Yes/no	40.0/60.0	40.3/60.0	>0.999 ^d^
Physical symptoms, mean (SD)			
MHR-QOL vasomotor symptom score (0–6)	3.4 (2.3)	2.3 (2.1)	0.001 ^a^
Psychological symptoms, mean (SD)			
MHR-QOL insomnia symptom score (0–6)	3.7 (2.2)	2.3 (2.1)	<0.001 ^a^
Hospital Anxiety and Depression scale Anxiety subscale score (0–21)	9.2 (4.3)	7.8 (3.7)	0.029 ^a^
Hospital Anxiety and Depression scale Depression subscale score (0–21)	8.7 (4.4)	7.3 (3.7)	0.024 ^a^

^a^ Mann–Whitney test. ^b^ Chi-square test. ^c^ Unpaired *t*-test. ^d^ Fisher’s exact test. *N*, number; SD, standard deviation; MHR-QOL: Menopausal Health-Related Quality of Life Questionnaire.

**Table 2 nutrients-14-01226-t002:** Comparison of the daily intake of nutrients and the frequency of headache.

Characteristic	Headache		*p*-Value
	Frequent (*N* = 60)	Not So Frequent (*N* = 349)	
Nutrition, Mean (SD)			
Protein (%E)	15.5 (3.3)	15.8 (3.1)	0.244 ^a^
Animal protein (%E)	8.8 (3.3)	9.0 (3.1)	0.585 ^a^
Vegetable protein (%E)	6.7 (1.3)	6.8 (1.2)	0.511 ^b^
Carbohydrate (%E)	53.5 (10.1)	52.2 (8.6)	0.235 ^a^
Ash (g/1000 kcal/day)	10.7 (2.5)	10.6 (2.0)	0.815 ^a^
Sodium (mg/1000 kcal/day)	2405 (551)	2316 (478)	0.173 ^a^
Potassium (mg/1000 kcal/day)	1525 (588)	1594 (439)	0.054 ^a^
Calcium (mg/1000 kcal/day)	325 (127)	336 (112)	0.142 ^a^
Magnesium (mg/1000 kcal/day)	147 (41)	153 (35)	0.072 ^a^
Phosphorus (mg/1000 kcal/day)	589 (133)	608 (129)	0.174 ^a^
Iron (mg/1000 kcal/day)	4.6 (1.5)	4.7 (1.1)	0.218 ^a^
Zinc (mg/1000 kcal/day)	4.53 (0.80)	4.58 (0.74)	0.432 ^a^
Copper (mg/1000 kcal/day)	0.65 (0.14)	0.66 (0.12)	0.344 ^a^
Manganese (mg/1000 kcal/day)	1.79 (0.64)	1.78 (0.61)	0.909 ^a^
Fat (%E)	27.2 (6.1)	28.0 (5.8)	0.330 ^b^
Animal fat (%E)	12.4 (4.5)	12.8 (4.2)	0.496 ^a^
Vegetable fat (%E)	14.8 (4.0)	15.2 (4.0)	0.500 ^b^
Saturated fatty acid (%E)	7.4 (2.1)	7.7 (1.9)	0.095 ^a^
Monounsaturated fatty acid (%E)	9.6 (2.4)	9.9 (2.3)	0.224 ^b^
Polyunsaturated fatty acid (%E)	6.6 (1.7)	6.7 (1.6)	0.350 ^a^
Cholesterol (mg/1000 kcal/day)	201 (74)	208 (73)	0.320 ^a^
N-3 fatty acid (%E)	1.3 (0.5)	1.4 (0.4)	0.295 ^a^
N-6 fatty acid (%E)	5.2 (1.4)	5.3 (1.3)	0.399 ^a^
Soluble dietary fiber (g/1000 kcal/day)	1.86 (0.81)	1.99 (0.66)	0.054 ^a^
Insoluble dietary fiber (g/1000 kcal/day)	5.18 (2.46)	5.38 (1.65)	0.064 ^a^
Dietary fiber (g/1000 kcal/day)	7.31 (3.56)	7.59 (2.37)	0.065 ^a^
Daidzein (mg/1000 kcal/day)	7.7 (5.8)	9.2 (5.4)	0.009 ^a^
Genistein (mg/1000 kcal/day)	13.0 (9.9)	15.6 (9.1)	0.009 ^a^
Isoflavone (mg/1000 kcal/day)	20.7 (15.7)	24.8 (14.5)	0.009 ^a^
Retinol (μg/1000 kcal/day)	254 (171)	223 (147)	0.360 ^a^
β-carotene (μg/1000 kcal/day)	2495 (2492)	2459 (1522)	0.108 ^a^
Retinol equivalent (μg/1000 kcal/day)	463 (262)	430 (202)	0.779 ^a^
Vitamin D (μg/1000 kcal/day)	6.92 (4.87)	7.29 (4.75)	0.294 ^a^
α-tocopherol (mg/1000 kcal/day)	4.45 (1.60)	4.42 (1.15)	0.417 ^a^
Vitamin K (μg/1000 kcal/day)	198 (149)	209 (101)	0.044 ^a^
Vitamin B1 (mg/1000 kcal/day)	0.45 (0.13)	0.46 (0.10)	0.183 ^a^
Vitamin B2 (mg/1000 kcal/day)	0.77 (0.23)	0.78 (0.20)	0.256 ^a^
Niacin (mgNE/1000 kcal/day)	9.64 (3.06)	9.85 (2.68)	0.425 ^a^
Vitamin B6 (mg/1000 kcal/day)	0.72 (0.25)	0.75 (0.20)	0.068 ^a^
Vitamin B12 (μg/1000 kcal/day)	5.14 (3.06)	5.06 (2.57)	0.677 ^a^
Folic acid (μg/1000 kcal/day)	210 (104)	210 (76)	0.356 ^a^
Pantothenic acid (mg/1000 kcal/day)	3.67 (0.84)	3.82 (0.77)	0.081 ^b^
Vitamin C (mg/1000 kcal/day)	73.6 (41.7)	73.8 (34.0)	0.499 ^a^

^a^ Mann–Whitney test. ^b^ Unpaired *t*-test. %E, % energy.

**Table 3 nutrients-14-01226-t003:** Associations between the daily intake of isoflavone (mg/1000 kcal/day) and the frequency of headache.

	Nutrient	OR	95% CI	*p*-Value
Model 1	Vitamin K (μg/1000 kcal/day)	1.000	0.998–1.004	0.547
	Isoflavone (mg/1000 kcal/day)	0.976	0.952–1.000	0.048
Model 2	Vitamin K (μg/1000 kcal/day)	1.000	0.998–1.004	0.625
	Isoflavone (mg/1000 kcal/day)	0.974	0.950–0.999	0.036
	MHR-QOL VMS score (0–6)	1.160	1.012–1.329	0.033
	MHR-QOL insomnia score (0–6)	1.334	1.152–1.546	<0.001
	HADS depression subscale score (0–21)	1.067	0.967–1.177	0.199
	HADS anxiety subscale score (0–21)	0.994	0.897–1.101	0.902

OR, odds ratio; CI, confidence interval; MHR-QOL, Menopausal Health-Related Quality of Life Questionnaire; VMS, vasomotor symptom; HADS, Hospital Anxiety and Depression Scale. Model 1: Association between daily intake of isoflavone (mg/1000 kcal/day), vitamin K, and headache. Model 2: Multivariate logistic regression model adjusted for VMS, insomnia, depression, and anxiety scores.

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
