# Peer review of "The Inverse Correlation of Isoflavone Dietary Intake and Headache in Peri- and Post-Menopausal Women"

_nutrients, 2022, doi:10.3390/nu14061226_

Round 1

Reviewer 1 Report

In this study authors investigate the relationship between isoflavone intake and menopausal headache symptoms. However, the current version of the manuscript is very limited in scope.  It only focuses on amount relationship between isoflavone intake and menopausal headache symptoms in middle-aged women.  Additionally, no clear mechanism is provided to explain the how isoflavone intake is associated with improve headaches in middle-aged women.  Another major concern is how decline in isoflavone intake leads to headache? When cause of headache is still questionable.  Based on these elements, I do not recommend the acceptance of the paper at this stage. 

Additional Concerns: 

  1. What was the Inclusion and exclusion criteria for the participants?  Explain in detail. 
  2. All the participants with smoking history, alcohol consumption, were included. Do authors believe that any of these factors have influence in occurrence of headache? 
  3. As authors mentioned that questionnaire only obtained the frequency of headaches without distinction between different types of headaches, such as migraine and tension headache. How authors confirm whether there is direct relation between headaches and isoflavone intakes? 
  4. Why authors combine both perimenopausal and postmenopausal women while comparing premenopausal women (Fig 1)? 
  5. Why authors did not show the steroid hormones level in the participants included in the study? 
  6. Headache can be due several factor eg, stress, anxiety, tension, amount of fluid intake, poor sleep and food habit, working style. Just correlating the dietary intake of isoflavone and headache in peri- and post-menopausal women and concluding that it may improve headaches in middle-aged women will be a little extrapolation of findings. 
  7. Table 1. : Is there a typological error in Table 1? Percentage of Menopausal status, perimenopausal/postmenopausal is same in both  headache (N = p value 60) and Not so frequent (N = 349) group. Menopausal status, perimenopausal/postmenopausal (%)- 32.4/23.1/44.5, 32.4/23.1/44.5
  8. Isoflavones is part of dietary supplement and suggesting that active consumption of isoflavones may improve headaches in middle-aged women' has not been justified in the manuscript.

Reviewer 2 Report

The project of the work is interesting, although in my opinion some issues require clarification:
1. Introduction a paragraph should be included explaining the choice of isoflavones and the mechanism of action as phytoestrogens. For a reader unfamiliar with this mechanism, the choice of authors is not clear
2. in the methodology chapter, it should be explained what was dictated by the division of the groups into frequent headaches (> 3) and not so frequent (<3). Moreover, the authors state that the qualified patients with tension headaches and migraines were qualified, and the mechanism of their formation is different, perhaps it would be worth distinguishing these groups, if not now, then in the future. Nevertheless, one would have to address this issue at work.

Round 2

Reviewer 1 Report

Authors have satisfactorily addressed most of my raised concerns.

Author Response

Thank you very much for a number of valuable comments in the first round, according to which the paper was revised and improved.